# Analysis of quantile regression for race time in standard distance triathlons

**Junhui Zhao**[1], **Yongfang Ma**[2]*, **Xiaoxiao Hu**[2]

**1** Department of Sports Teaching and Research, Lanzhou University, Gansu, China, **2** School of Mathematics and Statistics, Center for Data Science, Lanzhou University, Gansu, China

* 220220934740@lzu.edu.cn

**Data Availability Statement:** The race results data on triathletes is sourced from the Information System of the China Triathlon Sports Association. For further details please refer to the following link: http://triathlon.basts.com.cn/#game.

## Abstract

### Purpose

This study aims to quantitatively analyze the impact of split times on overall performance in standard distance triathlon events. It also examines how environmental factors such as water type, temperature, and altitude affect overall race outcomes.

### Methods

Quantile regression was employed to analyze the race records of 1,580 triathletes participating in 46 standard distance events in China.

### Results

Swim time significantly influences race performance among the top 50% of elite athletes (p < 0.05). For slower elite athletes, bike time is more critical. Temperature has a positive effect on race times, while altitude also shows a significant positive impact, with race times decreasing as altitude increases (up to 1,600 meters in this study's dataset). River water enhances race times compared to still water, whereas sea water generally slows athletes down.

### Conclusion

The influence of split times and environmental factors on overall race rime varies according to the athletes' performance levels. To optimize results, training plans and race strategies should be tailored to each athlete's capabilities. Additionally, understanding and adapting to environmental conditions in advance is crucial.

## Introduction

Triathlon is an endurance sport that involves sequential segments of swimming, transitioning from swimming to cycling (T1), cycling, transitioning from cycling to running (T2), and

**Funding:** This research was supported by the Fundamental Research Funds for the Central Universities (approval number: 21lzujbkytd009) and the Research Project of Gansu Provincial Department of Transportation, China: Mathematical Model Study on the Impact of Traffic Volume and Some Environmental Factors on the Performance of Ordinary National Highways in Gansu Province (Grants No. 2023-03). The funders had no role in study design, data collection and analysis, decision to publish, or preparation of the manuscript.

**Competing interests:** The authors have declared that no competing interests exist.

running, all over various distances [1]. Since becoming an Olympic sport, triathlon has evolved considerably [2].

Triathlon races are conducted over various distances, including sprint distance (750 m swimming, 20 km cycling, and 5 km running), standard distance (1.5 km swimming, 40 km cycling, and 10 km running), half-distance Ironman® triathlon (Ironman® 70.3) (1.9 km swimming, 90 km cycling, and 21.1 km running), full-distance Ironman® triathlon (3.8 km swimming, 180 km cycling, and 42.195 km running), and ultra-triathlons (which are longer or multiple times the full-distance Ironman®) [3–7]. In recent years, several studies have explored the relationship between the three split times and overall race time for triathlons across different distances. The cycling segment demonstrated the highest correlation between performance and overall time among elite male and female athletes in official sprint distance results [8].

Although the consistency between running performance and overall time was lower, it provided better explanatory power for determining overall rankings, particularly in elite men's and non-freestyle events [9]. For standard distance triathlons, the running segment is a critical predictive factor for overall time, while completing the swim in a split time close to the fastest is also important. Performance in the cycling leg is generally of lesser importance, although it holds more significance for women than for men [10]. In the full-distance Ironman® triathlon, the cycling split was the most crucial discipline [11, 12]. For age group triathletes competing in Ironman® 70.3, running and cycling are more predictive of overall race performance than swimming [13].

High-level athletes increasingly utilize altered environmental conditions, such as altitude and heat training, to optimize performance adaptations, supported by nutrition and hydration strategies to enhance their competitive capabilities [14]. The effective design of training programs, which includes individualized training plans and the integration of strength training, is crucial for optimizing athlete readiness and minimizing disruptions caused by health issues [15]. Furthermore, understanding the dynamics of training transfer, including factors such as self-efficacy and feedback, is essential for translating training into improved performance in competition [16]. Recent research has also highlighted the variability in performance between indoor and outdoor cycling environments, revealing that individual training history and body mass index (BMI) can significantly influence outcomes [17]. This underscores the necessity of tailoring training strategies to each athlete's specific context to maximize their competitive edge.

Additionally, studies have indicated that environmental factors can significantly impact triathlon pacing, including water currents, wind conditions, topography, ambient heat, and humidity during Ironman triathlons [18–21]. Competing in triathlons and ultra-endurance events in tropical climates presents various challenges, particularly concerning the effects of heat on the swimming, cycling, and running segments of the race [22]. Furthermore, ozone and PM2.5 concentrations have been shown to negatively affect the performance times of Ironman triathletes [23]. However, there is a scarcity of empirical analyses examining the effects of environmental factors on performance using data. Studies should consider the role of course characteristics and environmental conditions in the relationship between overall and split times in triathletes [24].

The study aims to investigate the influence of swim, bike, and run split times on overall time in standard distance triathlon events, as well as to analyze how specific environmental conditions affect overall time. In contrast to prior research, we have chosen to utilize race performance data from the Chinese triathlon field, intending to enrich the body of research on triathlon events and provide empirical evidence from races in China. We analyzed race results from 46 standard distance triathlon events held in China, encompassing elite male and female

athletes from both domestic and international backgrounds. Additionally, we incorporated race course environmental data and aimed to quantitatively assess the impact of these environmental factors on triathlon performance, providing valuable insights.

## Data collection and methods

### Data collection

This study focuses on the race results of 46 standard distance triathlon (Table 1) races held in China between 2013 and 2023. The race results data on triathletes is collected from the Information System of the China Triathlon Sports Association website (http://triathlon.basts.com.cn/#game) using a Python script. The collected data included various attributes such as the sex, name, swim, run, bike, and overall race times of the triathletes.

Specifically, the selected data includes a total of 1580 athletes from the Elite Group, consisting of 1007 elite men and 600 elite women, both Chinese and foreign. This group comprises both male and female elite athletes, corresponding to the Elite Men and Elite Women categories on the official World Triathlon website. Among these races, 19 were international competitions, such as the 2018 Taizhou ASTC Triathlon Asian Cup and the 2023 World Triathlon Cup Weihai, while 33 were domestic races, including the 14th National Games Triathlon and the Xiamen National Triathlon Championship in 2022. The official data for international races can also be found on the International Triathlon Union (ITU) website. It is important to note that the competition rules are the same for both international and domestic races, ensuring consistency in how events are conducted. For further details, reference should be made to the World Triathlon Competition Rules.

**Table 1. List of triathlon events analyzed.**

| Year | Race | Year | Race |
|---|---|---|---|
| 2013 | Chengdu ITU Triathlon Premium Asian Cup | 2018 | Weihai ITU Triathlon World Cup |
| 2014 | Chengdu ITU Triathlon World Cup | 2019 | Henan Suixian National Triathlon Championships |
| 2014 | Jiayuguan ITU Triathlon World Cup | 2019 | Jiayuguan National Triathlon Championships |
| 2014 | Shizuishan ASTC Triathlon Premium Asian Cup | 2019 | Jiangxi Dexing National Triathlon Championships |
| 2014 | Suzhou Wujiang National Triathlon Championships | 2019 | Dexing ASTC Triathlon Asian Cup |
| 2014 | Zhenjiang ASTC Triathlon Premium Asian Cup | 2019 | Lianyungang ASTC Triathlon Asian Cup |
| 2015 | Harbin National Triathlon Championships | 2019 | Shantou Triathlon Championships |
| 2015 | Henan Suixian National Triathlon Championships | 2019 | Shantou ASTC Triathlon Asian Cup |
| 2015 | Shizuishan ASTC Triathlon Premium Asian Cup | 2019 | Weihai ITU Triathlon World Cup |
| 2015 | Weihai ITU Long Distance Triathlon World Championships | 2021 | Dexing National Triathlon Championship Cup |
| 2015 | Changshou ASTC Triathlon Premium Asian Cup | 2021 | The 14th National Games Triathlon Competition |
| 2016 | Ningxia Shizuishan National Triathlon Championship Cup | 2021 | Huaian National Triathlon Championships |
| 2016 | Taizhou ASTC Triathlon Asian Cup | 2021 | Lianyungang National Triathlon Championship |
| 2016 | Chongqing Changshou Lake National Triathlon Championships | 2022 | Dongying National Triathlon Championship Cup Series |
| 2017 | China Triathlon League—Suixian, Henan Province | 2022 | Harbin National Triathlon Championship Cup Series |
| 2017 | China Triathlon League—Taizhou, Jiangsu Province | 2022 | Lianyungang National Triathlon Champion Cup Series |
| 2017 | China Triathlon League—Beidaihe, Qinhuangdao | 2022 | Xiamen National Triathlon Championships |
| 2017 | China Triathlon League-Suidong Dongdaihe | 2023 | Dexing National Triathlon Championship Cup Series |
| 2017 | Triathlon Competition of the Thirteenth Games of the People's Republic of China | 2023 | Asia Triathlon Cup Dexing |
| 2018 | Henan Suixian National Triathlon Championship Cup | 2023 | Lianyungang National Triathlon Champion Cup Series |
| 2018 | Ningbo Dongqian Lake National Triathlon Championships | 2023 | Asia Triathlon Cup Lianyungang |
| 2018 | Dongdaihe ASTC Triathlon Asian Cup | 2023 | Taizhou National Triathlon Championship Cup Series |
| 2018 | Taizhou ITU Triathlon World Cup | 2023 | Asia Triathlon Cup Taizhou |

Moreover, environmental indicators, such as temperature, elevation, and waters, are obtained from the official website (https://lishi.tianqi.com/). The temperature represents the highest temperature recorded at the race venue on the day of the local event, while elevation refers to the elevation of the event venue above sea level. Waters categorization includes still water, seawater, and river water, which are typically used in the swimming events of a triathlon. Still water refers to bodies of water, such as lakes or reservoirs, where there is minimal or no current. River water, on the other hand, is characterized by its flowing nature due to the force of the current. Seawater refers to the saltwater found in oceans and seas. Triathlons held in coastal areas often involve swimming in seawater.

## Data treatment

Results with incomplete records are excluded based on specific criteria. This includes triathletes who did not start or finish, disqualified triathletes, individuals with missing split times, and inconsistent time records. Subsequently, the data records are preprocessed by converting them from the hh:mm:ss format to raw times in seconds.

## Statistical analysis

### Model building

We employed quantile regression (QR) methodology, originally developed by Koenker and Bassett [25, 26], to gain deeper insights into the effects of independent variables on the conditional distribution of the dependent variable, specifically overall triathlon time. Unlike traditional regression analysis, which primarily focuses on mean values, quantile regression examines the impact of explanatory variables across different quantiles of overall time (Y values). This analytical approach facilitates an in-depth exploration of how specific sub-event times—such as swimming, cycling, and running—along with external factors like temperature, elevation, and water conditions, influence overall results.

The selection of quantile regression is particularly relevant because the relationships between sub-event times and environmental factors may not be linear and can vary depending on the level of overall performance. By employing a quantile regression model, we can investigate how the effects of these variables differ across various total time levels in triathlons. This allows for a more nuanced understanding of the dynamics involved, as the influence of specific sub-event times and external conditions may shift based on the performance quartile of the athletes.

In our study, the dependent variable is overall time, which serves as an indicator of the triathlete's performance. While a shorter race completion time does not directly equate to better performance, it can still reflect the athlete's level of ability in the context of triathlons to some extent. The independent variables in our model include swim time, bike time, run time, temperature, elevation, and water type (which is converted into a dummy variable with still water as the reference category).

A quantile regression model is created as follows:

$$Y = \alpha_\tau X_1 + \beta_\tau X_2 + \delta_\tau X_3 + \gamma_\tau X_4 + \varphi_\tau X_5 + \theta_\tau X_6 + \varepsilon \tag{1}$$

where $Y$ is the overall time, $X_1$, $X_2$ and $X_3$ represent swim time, bike time and run time, $X_4$, $X_5$, $X_6$ represent temperature, elevation, and water type, respectively.

### Statistical analysis

Before conducting any analyses, we first presented the descriptive statistics for the variables, including means, standard deviations, and ranges. To quantitatively assess the impact of

**Table 2. Descriptive statistics for numerical variables.**

| Variable | Count | Mean | SD | Min | Max | Unit |
|---|---|---|---|---|---|---|
| overall time | 1,580 | 7412 | 689.1143 | 6179 | 10722 | s |
| swim time | 1,580 | 1181 | 78.04042 | 1040 | 1642 | s |
| run time | 1,580 | 2257 | 251.3733 | 1746 | 3302 | s |
| bike time | 1,580 | 3857 | 460.4429 | 3175 | 6151 | s |
| temperature | 1,580 | 25 | 4.437 | 14 | 35 | ˚C |
| elevation | 1,580 | 128 | 273.6958 | 0 | 1600 | m |

environmental factors, we employed quantile regression analysis to estimate the effects of temperature, elevation, and water type across different quantile points. This method allowed us to simultaneously evaluate the relationship between split times and overall race times. Statistical significance was determined at a threshold of $p < 0.05$. All statistical analyses were performed using Microsoft Excel 2016 and Stata 16 software.

## Results

### Descriptive statistics for variables

Table 2 presents descriptive statistics for the exercise results variables, including total time, segment times for swim, bike and run, and also descriptions of environmental factors such as temperature and elevation.

Table 3 presents descriptive statistics for categorical variables, including the percentage of athletes by sex and the percentage of different water categories (still water, river water and seawater).

### Quantile regression models

Regression analysis was conducted separately for the 0.1, 0.5, and 0.9 quantiles. The results are presented in Table 4 (Quantile regression results of male triathletes) and Table 5 (Quantile regression results of female triathletes).

The coefficients for all three sub-disciplines show p-values less than 0.05, indicating statistical significance. In both the 0.1 and 0.5 quantile regression results, swim time has the largest coefficient, highlighting its substantial impact on overall times (in seconds) for both male and female athletes. Conversely, the 0.9 quantile regression results reveal a shift in the coefficients, with bike time exhibiting the largest coefficient and swim time the smallest. This suggests that at lower achievement levels (longer overall race times), bike time becomes a more critical factor in determining overall performance.

**Table 3. Descriptive statistics for categorical variables.**

| Variable | Classification | Count | Percentage |
|---|---|---|---|
| gender | male | 1007 | 63.7% |
| | female | 573 | 36.3% |
| total | —— | 1580 | 100% |
| waters | still water | 898 | 56.8% |
| | river water | 120 | 7.6% |
| | seawater | 562 | 35.6% |
| total | —— | 1580 | 100% |

**Table 4. Quantile regression results of male triathletes.**

| Variables | 0.1 | 0.5 | 0.9 |
|---|---|---|---|
| Swim time | 1.031*** | 1.092*** | 0.998*** |
| | (0.000) | (0.000) | (0.000) |
| Bike time | 1.005*** | 1.055*** | 1.032*** |
| | (0.000) | (0.000) | (0.000) |
| Run time | 1.010*** | 1.003*** | 1.023*** |
| | (0.000) | (0.000) | (0.000) |
| Temperature | -2.559*** | -1.957*** | -3.707*** |
| | (0.000) | (0.000) | (0.000) |
| Elevation | -0.004*** | -0.013 | -0.039*** |
| | (0.000) | (0.063) | (0.000) |
| River water | -12.009*** | -17.379*** | -41.425*** |
| | (0.000) | (0.000) | (0.000) |
| Sea water | 6.281*** | 6.543 | 27.203*** |
| | (0.000) | (0.091) | (0.000) |
| Constant | 68.014*** | -155.199*** | 84.317*** |
| | (0.000) | (0.000) | (0.002) |

Note

\* \* \*, \* \* represent 1%, 5% level of significance respectively

Specifically, at the 0.1 and 0.5 quantiles, the coefficients for the three sub-disciplines rank as follows: swim time has the highest coefficient, followed by bike time, with run time having the lowest influence on overall outcomes. As the quantile increases from 0.1 to 0.5 and 0.9, the coefficient for run time also increases. At the 0.9 quantile, the ranking shifts to bike time, run time, and swim time, suggesting that run time has an enhanced impact at this level.

**Table 5. Quantile regression results of female triathletes.**

| Variables | 0.1th | 0.5th | 0.9th |
|---|---|---|---|
| Swim time | 1.047*** | 1.080*** | 0.857*** |
| | 0.000 | 0.000 | 0.000 |
| Bike time | 1.018*** | 1.054*** | 1.074*** |
| | 0.000 | 0.000 | 0.000 |
| Run time | 0.999*** | 1.006*** | 1.009*** |
| | 0.000 | 0.000 | 0.000 |
| Temperature | -2.111*** | -1.220*** | -4.539*** |
| | 0.000 | 0.001 | 0.000 |
| Elevation | -0.008 | -0.030*** | -0.053*** |
| | 0.108 | 0.000 | 0.000 |
| River water | -16.170*** | -28.297*** | -60.320*** |
| | 0.001 | 0.000 | 0.000 |
| Sea water | -3.837** | -28.237*** | -26.471*** |
| | 0.042 | 0.000 | 0.000 |
| Constant | 8.879 | -171.172*** | 152.570*** |
| | 0.775 | 0.000 | 0.013 |

Note

\* \* \*, \* \* represent 1%, 5% level of significance respectively

The coefficients for temperature consistently yield p-values less than 0.05, indicating a significant effect. The negative coefficient suggests that an increase in temperature is associated with a decrease in overall time (in seconds).

For elevation, the coefficients at the 0.1 quantile for males (p-value = 0.063) and the 0.5 quantile for females (p-value = 0.108) exceed 0.05, indicating non-significance. However, in other quantiles, the coefficients are negative, implying that increased elevation is associated with a decrease in overall time (in seconds).

The coefficients for river water consistently show p-values less than 0.05, indicating a significant impact. The negative coefficient demonstrates that overall time (in seconds) is reduced in river water environments compared to still water. The coefficients for sea water yield p-values less than 0.05 in most analyses, except for the male 0.5 quantile and female 0.1 quantile, where the p-values exceed 0.05, indicating non-significance. The positive coefficient for sea water suggests that overall time (in seconds) increases in sea water environments compared to still water.

In summary, the quantile regression analysis reveals that various environmental factors and sub-disciplines impact the overall times of triathletes differently across different quantiles. The findings indicate that swimming is paramount at higher performance levels, while cycling becomes more significant at lower performance levels.

Additionally, environmental factors such as temperature, elevation, and water conditions substantially influence overall race times.

## Discussion

### Impact of split times

The coefficients for swim time being higher than those for bike and run times at the 0.1 and 0.5 quantiles indicate that, for athletes competing at this level, swim time significantly impacts overall race performance. This finding supports the conclusions of Peeling and Landers (2009), who noted that finishing the swim leg in the "first pack" is crucial for achieving success [27]. An analysis of standard triathlons suggests that competitive athletes should aim to complete each segment as quickly as possible, especially ensuring they are in the "first pack" from the swimming split [10]. The overall finishing position of elite triathletes in the standard distance triathlon correlates significantly with average swimming velocity and the athlete's position after the swim stage [28]. Furthermore, Landers et al. (2008) found that winners emerged from the water in the first pack in 90% of elite male and 70% of elite female races [29]. Swimming serves as a better predictor of overall performance in Olympic-distance triathlons, as strong swimmers have a higher likelihood of success for two main reasons: (1) swimming comprises a larger proportion of the race compared to longer distances (e.g., Ironman 70.3 and Ironman 140.6) and (2) faster swim times allow athletes to position themselves within a faster cycling peloton [13].

The 0.9 quantile regression results reveal a shift in the coefficients, with bike time exhibiting the largest coefficient and swim time the smallest. A potential reason for the observed changes in the impact of each segment on overall performance could be the distribution of energy and the accumulation of fatigue throughout the race. Thus, athletes at this competitive level should prioritize enhancing their performances in both the bike and run segments. By strategically allocating their energy and focusing on reducing times in these two disciplines, athletes can significantly lower their overall race time and improve their chances of achieving better results and higher rankings.

However, there is currently a lack of research specifically addressing how the influence of these three disciplines varies across different performance levels. One certainty is that the

impact of each segment changes based on the athlete's proficiency, highlighting the need for tailored training strategies.

## Impact of environmental factors

**Temperature.** The negative coefficient indicates that, holding other conditions constant, an increase in temperature is associated with a decrease in overall race time. Typically, high temperatures negatively affect athletic performance. Hot and humid climates impact performance and withdrawal rates, as observed in the Kona Ironman World Championships held every October [22, 30]. Top age-groupers were slower in Hawaii than in their qualifying races, with an abandon rate that can reach 10%, which is significant considering the fitness level of athletes participating in this particular event, who are generally more resistant to heat stress [31].

This unexpected outcome can be attributed to several factors. One primary reason is related to the characteristics of the dataset used in this study. The analysis utilized the highest recorded temperatures on race days, but actual environmental conditions likely fluctuated and did not consistently reflect these peak temperatures. The temperature range across events varied from 14°C to 35°C, with an average of 25°C. Notably, only one event reached 35°C (the 2021 Lianyungang National Ironman Championship) and another reached 33°C (the 2021 Huai'an National Ironman Championship). The quantile regression results did not adequately address how triathlete performance is impacted by extremely high temperatures ($\geq$35°C), as existing literature predominantly identifies such conditions as detrimental to endurance performance.

Another factor may be the interplay between air temperature and water temperature. Typically, air temperature closely correlates with water temperature; lower air temperatures can lead to colder water, presenting challenges for athletes during the swimming phase. Research indicates that at water temperatures of 25°C and 18°C, average oxygen consumption during arm and leg ergometry increases by 9% and 25.3%, respectively, compared to swimming in 33°C water [32, 33]. Conversely, warmer ambient temperatures often correlate with warmer water, enhancing athletes' physical readiness at the start of a race, which can improve performance during the initial swimming segment and positively influence overall race time.

Furthermore, various interventions have been shown to mitigate the adverse effects of heat, including heat acclimation/acclimatization, physical training, pre-exercise cooling, and fluid ingestion [34]. These strategies effectively reduce thermoregulatory strain and enhance endurance performance in warm conditions. Consequently, athletes competing under these circumstances may have developed effective coping mechanisms that allow them to perform optimally despite the heat, resulting in an overall reduction in total race time.

In conclusion, while the negative correlation between temperature and overall race time may initially appear counterintuitive, it underscores the complexity of endurance performance dynamics. Further research is warranted to explore these relationships, particularly under extreme temperature conditions, to gain deeper insights into the factors influencing triathlete performance.

**Elevation.** The elevation range in the analyzed events spans from 0 to 1,600 meters, with an average elevation of just 128 meters. Notably, the only event that occurred at an elevation of 1,600 meters was the 2019 Jiayuguan National Triathlon Championships and U-Series Championships; all other race courses featured elevations below 1,000 meters. The coefficients for elevation demonstrate a negative correlation with overall time, suggesting that as elevation increases, completion times (measured in seconds) become shorter.

A notable historical example of elevation's impact on athletic performance occurred during the 1968 Olympic Games in Mexico City, held at an elevation of 2,340 meters [35, 36]. World

records were set in all track events up to 800 meters, including the 100 meters, 200 meters, 400 meters, 4x100 meter relay, and 4x400 meter relay. Increased elevation reduces air density (~1% for every 100 m [37]), which affects aerodynamic drag and facilitates high-speed movements (e.g., running [38], speed skating [39]), while decreasing the energy cost of running at high speeds without diminishing energy availability [40].

In addition to this historical perspective, research conducted between 2000 and 2009 revealed marginal improvements of approximately 0.2% in elite athletes competing in sprint events (ranging from 100 to 400 meters) at terrestrial elevation of 500–999 meters. Similarly, performances at elevation of 1000 meters or higher exhibited faster completion times, with improvements ranging from 0.1% to 0.5% in events such as the 100-meter sprint to the 400-meter hurdles.

This reduction in drag can be beneficial for athletic performance, as highlighted in the study titled "Making History in 1 h: How Sex, Aging, Technology, and Elevation Affect the Cycling Hour Record." This research shows that elevation can positively impact performance up to 1,000 meters; beyond this point, the advantages begin to diminish due to declining aerobic performance that eventually outweighs the benefits from reduced aerodynamic drag [41].

**Waters.** Open-water swimming (OWS) has surged in popularity in recent decades; however, the existing literature remains relatively sparse, especially regarding the performance dynamics of elite and age-group athletes [42]. This gap is significant, given the unique environmental characteristics that define open-water races, such as variations in water temperature, tides, currents, and wave action [43]. These factors can greatly influence athletes' performance by affecting their tactical decisions and pacing strategies.

Our regression analysis reveals that athletes competing in river environments achieve a statistically significant reduction in overall completion time compared to those swimming in still water. Conversely, our findings indicate that seawater environments typically correlate with increased overall race times. This suggests that the density, buoyancy, and waves of seawater present specific challenges that can hinder performance. Nonetheless, the underlying mechanisms by which river and seawater affect athletic performance require further exploration.

These findings underscore the critical need for triathletes and coaches to prioritize open-water training to optimize performance under various conditions. Emphasizing this type of training is essential for improving swimming efficiency (SI), which is vital for success in competitive events [44]. By adapting to the unique demands of open-water environments through targeted practice, athletes can enhance their performance and competitive edge.

**Limitations.** We acknowledge that a limitation of this study is the potential endogeneity arising from using split times as explanatory variables while estimating their impacts on total race times. We recognize that this could introduce bias into our results. Despite our efforts to mitigate this issue, including the exploration of instrumental variables, we were unable to identify a suitable one within the current dataset. Therefore, this issue warrants further investigation in future research.

Additionally, this paper did not analyze the effect of water precooling. The model considered in this study could be enhanced by incorporating more environmental factors, such as wind speed, terrain, and water temperature. However, accessing relevant data is challenging, as these variables are often under-recorded and not readily available on official websites. Other researchers have also encountered limitations in collecting data on high-level athletes, including restricted sample sizes, difficulties in coordinating competition schedules with measurements, challenges in data acquisition, and concerns regarding data quality and applicability [45].

Future studies could focus on analyzing internationally recognized competitions to gather comprehensive environmental data from standard courses. Conducting an in-depth analysis

of this data would provide valuable insights into the specific impacts of environmental conditions on triathletes' performance. Such analysis would enable the customization of training regimens to better prepare triathletes for these conditions.

## Conclusions

This study highlights the varying impacts of swim, bike, and run times on overall race performance among triathletes at different competitive levels. Quantile regression analysis reveals that swim time has the greatest impact for the top 50% of elite athletes, underscoring the importance of finishing the swim leg in a strong position. In contrast, for athletes with longer overall race times, bike and run performances become more critical, suggesting that training priorities should shift accordingly.

Environmental factors, including temperature, elevation, and water conditions, significantly influence triathlete performance. These findings emphasize the necessity for athletes to adapt their training strategies to their performance levels and the specific environmental conditions they will face in their races.

Future research should explore additional environmental variables and their mechanisms of influence on athletes at varying performance levels. This understanding could further inform targeted training strategies, helping athletes optimize their preparation and pacing decisions based on anticipated race conditions. Coaches and athletes can leverage these insights to design effective training programs focused on improving specific split times, ultimately enhancing overall race performance in standard distance triathlons. By considering environmental factors, triathletes can make informed choices regarding equipment and pacing strategies, leading to improved race outcomes.

## Author Contributions

**Conceptualization:** Junhui Zhao.

**Formal analysis:** Yongfang Ma.

**Funding acquisition:** Junhui Zhao, Xiaoxiao Hu.

**Methodology:** Junhui Zhao, Yongfang Ma, Xiaoxiao Hu.

**Resources:** Junhui Zhao, Xiaoxiao Hu.

**Supervision:** Xiaoxiao Hu.

**Writing – original draft:** Yongfang Ma.

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
