## [Decision Letter · Decision Letter 0]

6 May 2024

PONE-D-24-09996The Impact of Split Times and Environmental Factors on Performance in Olympic Distance Triathlons: A Quantile Regression AnalysisPLOS ONE

Dear Dr. Zhao,

Thank you for submitting your manuscript to PLOS ONE. After careful consideration, we feel that it has merit but does not fully meet PLOS ONE’s publication criteria as it currently stands. Therefore, we invite you to submit a revised version of the manuscript that addresses the points raised during the review process.The overarching concern with the paper relates to the authors use, or perhaps misuse of certain statistical methods. As one of our reviewers is an expert in this area, I strongly encourage the authors to seeks additional counsel on their analyses to address these issues.Secondary to the statistical methodology employed, I recommend the authors, when appropriate, clearly articulate their rationale for specific modelling and interpretation within the text. Having reviewed the paper myself, I concur with reviewer one regarding the authors (lack of) discussion and conclusions, especially considering the lack of supporting references, for their interpretation of environmental impacts on performance. As noted, the authors conclusions are counter-intuitive and lack supporting evidence. Some of my specific concerns include:There is wide and consistent evidence that running times slow above 10, 15, and certainly 20d C, so it seems implausible that running would be positively affected at 30-35d C. Did the authors account for a pre-cooling effect of water on performance in warmer temperatures?Similarly, the assumption that athletes perform better at altitude because many train at altitude makes little sense. There is a consistent steady decline in performance from 1500m onward regardless of acclimation; ie, you’re pretty much always lower at altitude. Swimming at altitude offers no advantage that I know of and for the unacclimated, a greater hypoxic challenge. Any aerodynamic gains made at altitude are complex and only plausible in cycling where speeds are high enough to benefit from those specific gains; ie, speed is high enough. In either case, the authors need to provide some greater explanation for such gains beyond just the model.You may gain some insight on how this was handled from a recent paper I co-authored. Rather than a citation, it may aid in how you approach explaining the performance seen that your models suggest. DOI: 10.1249/MSS.0000000000003328 Please submit your revised manuscript by Jun 20 2024 11:59PM. If you will need more time than this to complete your revisions, please reply to this message or contact the journal office at plosone@plos.org. Please include the following items when submitting your revised manuscript:A rebuttal letter that responds to each point raised by the academic editor and reviewer(s). You should upload this letter as a separate file labeled 'Response to Reviewers'.A marked-up copy of your manuscript that highlights changes made to the original version. You should upload this as a separate file labeled 'Revised Manuscript with Track Changes'.An unmarked version of your revised paper without tracked changes. You should upload this as a separate file labeled 'Manuscript'.

We look forward to receiving your revised manuscript.

Kind regards,

Chris Harnish, PhD

Academic Editor

PLOS ONE

Journal Requirements:

"This work was supported by the Fundamental Research Funds for the Central Univer-sities (approval number: 21lzujbkytd009)."

"The authors would like to express their gratitude for the financial support provided through the Fundamental Research Funds for the Central Universities (approval number: 21lzujbkytd009)."

"This work was supported by the Fundamental Research Funds for the Central Univer-sities (approval number: 21lzujbkytd009)."

"No potential conflict of interest was reported by the author(s)"

5. We notice that your supplementary [S1-S6 Fig] are included in the manuscript file. Please remove them and upload them with the file type 'Supporting Information'. Please ensure that each Supporting Information file has a legend listed in the manuscript after the references list.

Reviewers' comments:

Reviewer's Responses to Questions

**Comments to the Author**

1. Is the manuscript technically sound, and do the data support the conclusions?

Reviewer #1: Partly

Reviewer #2: No

2. Has the statistical analysis been performed appropriately and rigorously? 

Reviewer #1: No

Reviewer #2: No

3. Have the authors made all data underlying the findings in their manuscript fully available?

Reviewer #1: Yes

Reviewer #2: No

4. Is the manuscript presented in an intelligible fashion and written in standard English?

Reviewer #1: Yes

Reviewer #2: No

5. Review Comments to the Author

Reviewer #1: The paper undertakes an interesting start, uses an interesting dataset, and asks interesting questions using statistical techniques. However, I believe the paper suffers from both serious and not so serious flaws in the statistical methods. The most serious flaw is that the paper seemingly tries to estimate impacts of splits on total times and then uses split times as right hand side/explanatory variables. This is strictly a nonstarter as these are endogenous regressors. The paper employs quantile regression instead of OLS, without ample reasoning why; I can only speculate but I'd suspect that OLS failed with perfect explanatory power of the three split times on the RHS. Not surprisingly, then, the paper finds coefficients near unity for the split times, and then interprets these coefficients, with sometimes counterintuitive results (not surprisingly). The conclusions of the paper are then based on this. Smaller statistical method problems also come from a poor explanation of why quantile regression is better and, in this case, should be preferred over OLS or other linear methods. The paper also compares some group mean differences and uses some sophisticated methods based on a test of normality of the outcome variables, but fails to explain a) WHY this variable is expected to be normal, b) why normality is required, and c) why a straightforward comparison of sample means or other sample statistics, with known normal or t-stat sampling distributions, cannot be used for comparison including sampling statistical variation. Start with simple statistics and then build up. Normality is not even required for OLS to be Best Linear Unbiased (Gauss Markov Theorem)--only for the correct standard error estimates.

Finally, some of the conclusions are quite counter intuitive, such as altitude increasing speed. This might be the case for cycling, but running and swimming? These I would presume would change, but in any case such as this a further discussion and in depth comparison with intuition, theory and prior research would help lead reader and author to sound advancing of the science.

The paper has some chance for redemption, as this rich dataset covers all kinds of things such as temperature, altitude, gender, age, and split times. It covers a signficant time period and interestingly covers Asia and China and other countries. the chance to include country, specific event quality, and other effects also exists. Many ways about this could be done -- for example, either IV methods could be considered or other modelling options, such as using share equations (i.e., dividing through by total time and estimating a system of share equations) could be explored. The current author might consider a co-author with a background in econometrics and/or statistical modelling.

Reviewer #2: In the present study, the authors investigated the possible effect of split times and also environmental factors on the performance of the Olympic distance triathletes. Despite the interesting nature of the study, I regret to say that there are some critical points and considerations regarding the writing style, logic and coherence of the introduction, statistics, results section and also discussion. I put some comments to support my decision of not accepting the manuscript in the current format.

1. The title of the manuscript is in a way as if this study is cause-and-effect study while there is no intervention and the authors just investigated possible relationship or contribution of some factors to performance in Olympic distance triathlons. It’s misleading and I recommend the authors to consider this point.

2. Line 47, there is no logical connection between the sentence “This study also focuses…” and the previous parts of the introduction.

3. The introduction does not follow a logical and coherent approach to introduce the main issue, present the current knowledge, indicate the gaps and novel aspects of the study, and finally form a concrete hypotheses or questions. I highly recommend the authors to reconstruct the introduction.

4. Line 95, please include the name of the official website form where the environmental data were collected.

5. Line 106, what the authors means by “inconsistent time record” which was set as an exclusion criterion?

6. In Table 2, there is an inconsistency between total time and the time of each segment (swim, bike, and run).

7. Line 108, the Statistical Analysis section must include all the details related to any statistical tests used in the study. There are some missing points that should be considered.

8. Is investigating the difference between two sexes one of the study goals? Nothing has been mentioned in this regard.

9. While most of the quantile regression studies are interested in finding the median of Y, why the authors set the Ƭ of 0.1th and 0.9th in the present study?

10. In the results section, more detailed information regarding the order of influence of each independent variables for each quintile and also for sexes must be provided. All these information is missing in the results section. In the meantime, the authors have used such information in the discussion section without presenting them in the result section.

11. In the discussion section, there are no strong support for the findings of the study. For example, the authors arguing that by increasing the altitude the performance increases and attributed this finding to acclimatization of the athletes. However, the mean altitude of races was 186 m above the sea level which normally has no considerable effect of body. In another example, the authors stated that the difference in the importance of running and swing as the predictors of the overall time is attributed to the difference in physical activity of athletes at 0.1 and 0.9 percentile. Why is that? No further information has been provided.

12. The main text of the manuscript needs a very careful revision in terms of the grammar, writing style, and verbs’ tense.

6. PLOS authors have the option to publish the peer review history of their article (what does this mean?). If published, this will include your full peer review and any attached files.

Reviewer #1: No

Reviewer #2: No

---

## [Author Response · Author response to Decision Letter 0]

8 Jul 2024

Response to reviewers

Dear Chris Harnish,

We appreciate you and the reviewers for dedicating your valuable time to reviewing our paper and providing valuable comments. Your insightful comments have led to potential improvements in the current version. The authors have carefully considered the comments and have made every effort to address each one of them. We hope that the manuscript, after careful revisions, meets our high standards. The authors welcome additional constructive comments, if there are any. All revisions are highlighted in blue color in the manuscript.

First of all, thank you for sharing the reviewer's comments with us. We appreciate the insightful feedback and the opportunity to address the concerns raised. We have addressed the concerns raised as follows: 

We have incorporated two new sections, "Methods Selection" and "Model Building," within the "Data" part of our manuscript to clearly articulate our rationale for specific modeling choices and interpretations. 

Some of these conclusions may seem counterintuitive, such as the positive effects of altitude and temperature. So, we have added more supporting literature in the discussion section, for example on lines 282-284. 

Pre-cooling with water is indeed an important aspect to be considered and we have reviewed the literature on this subject. However, the reason that water pre-cooling is not considered in the current study is that data on this aspect is not readily available. We have included this as a limiting factor on line 367 in the manuscript. Thank you for pointing this out. We will carefully consider this issue and investigate it in future studies.

We have carefully studied your recent paper co-authored with others (DOI: 10.1249/MSS.0000000000003328) to gain a deeper understanding of how to explain the outcomes suggested by our models. The paper titled "Making History in One Hour: How Gender, Age, Technique, and Altitude Affect Hour Record Cycling" delves into factors influencing the hour record cycling performance, and it deserves broader visibility and recognition for its valuable insights. We have cited this paper as it significantly contributes to explaining the models and results presented in our study. 

Once again, we appreciate the editor's guidance and support.

When submitting revision, we have addressed these additional requirements to meet Journal Requirements.

1. we ensure that our manuscript meets PLOS ONE's style requirements, including those for file naming. 

2.We have stated what role the funders took in the study: "The funders had no role in study design, data collection and analysis, decision to publish, or preparation of the manuscript." 

3.We have removed any funding-related text from the manuscript. The updated statement reads as follows: “This research was supported by the Fundamental Research Funds for the Central Universities (approval number: 21lzujbkytd009) and the Research Project of Gansu Provincial Department of Transportation, China: Mathematical Model Study on the Impact of Traffic Volume and Some Environmental Factors on the Performance of Ordinary National Highways in Gansu Province (Grants No. 2023-03).” 

4.We have declared that we have no competing interests.

5.We have removed the Supplementary Material [Figures S1-S6] and uploaded it separately as a “Supporting Information” file type, with a note following each Supporting Information file.

Please let us know if any further adjustments are required.

Below, the comments of the reviewers are addressed point by point, and the revisions are indicated.

Response to Reviewer #1

Thank you very much for reviewing our paper and your valuable suggestions. The issues you raised are very important to us and we will try our best to explain and improve our research. Regarding the serious and less serious flaws in the statistical methodology of the paper that you have mentioned, we are willing to elaborate and explain each of them:

1. Thank you for highlighting this issue. Your question is invaluable, and we have endeavored to follow your suggestion and make attempts to address the problem. Despite our efforts to locate additional variable data, we have not yet identified highly suitable instrumental variables. 

Regarding your concern, after conducting quantile regression, we performed correlation tests between the residuals and three endogenous variables ((swim time, bike time, and run time). Based on the results from the regression at the 0.5 quantile for men, the correlations between the residuals and swim time, bike time, and run time were -0.017, -0.072, and 0.039, respectively, all below 0.1. This low correlation holds true across the 0.1, 0.5, and 0.9 quantiles. Therefore, despite the presence of endogeneity, its impact on the regression results is minimal, suggesting the current results are reasonably accurate.

Additionally, we have acknowledged this endogeneity issue as a limitation of our study in the discussion section. Moving forward, we remain committed to addressing this challenge by focusing on finding appropriate instrumental variables in our future research. Your input has been crucial in shaping our approach, and we welcome any further suggestions or considerations you may have.

2. The "Methods Selection" section provides a detailed explanation of why quantile regression was chosen as the analytical tool for exploring the performance of top-level triathletes. 

The reason we use quantile regression is that we want to focus on the performance of athletes at different levels through quantile regression. For example, the effect of the three split times versus the overall time may be different depending on the level of the athlete. Quantile regression allows for a more comprehensive analysis of the conditional distribution of the dependent variable, which provides insight into how the independent variable affects different parts of the distribution and can capture potentially nonlinear relationships (three split times with overall time). 

On the other hand, the effect of environmental factors on an athlete's performance can also be related to the level of the athlete, where high level athletes may be able to cope with the environment and maintain their performance, whereas lower-level triathletes may be subject to changes in their performance as a result of the environment. Athletes may have outliers in their performance due to factors such as the environment. Quantile regression handles these issues better than OLS.

In summary, the overall race time data exhibit right-skewness and non-normality, precluding the use of OLS regression. Moreover, our primary interest lies in the performance of athletes at the top 10th percentile level. Quantile regression addresses both these issues effectively, which is why we opted for this method.

3. The purpose of comparing the performance differences between male and female athletes was to determine if there is a significant difference. If there is a substantial difference, it may not be appropriate to treat them as a single population. Thus, subsequent experiments and discussions were conducted separately for males and females. 

We did not assume or expect the variables to follow a normal distribution. The normality test was performed to determine the appropriate method for testing the differences. The results indicated that the data did not follow a normal distribution, leading us to choose non-parametric tests to examine the performance differences between male and female athletes. If the data had met the assumption of normality, we would have used parametric tests suitable for normally distributed data. Therefore, the choice of non-parametric tests was not dependent on the assumption of normality. We started with the simplest statistical approach, considering the data did not follow a normal distribution. Hence, we did not utilize parametric methods such as t-tests or analysis of variance (ANOVA). Instead, we chose the commonly used non-parametric Mann-Whitney test to examine the performance differences between male and female athletes. Importantly, this test does not rely on the assumption of normality.

4. For the positive effects of temperature, we have added more supporting literature in the discussion section, for example, on lines 272-296. The influence of altitude on the performance of top 10% male and female triathletes is not significant. However, for male and female triathletes at the 0.9 quantile, we have substantiated and explained the positive impact of altitude based on our dataset and relevant literature in the discussion section.

Response to Reviewer 2

Comment 1: The title of the manuscript is in a way as if this study is cause-and-effect study while there is no intervention and the authors just investigated possible relationship or contribution of some factors to performance in Olympic distance triathlons. It’s misleading and I recommend the authors to consider this point.

Our response: We gratefully appreciate for your valuable comment. We appreciate the reviewer's concern regarding the potentially misleading nature of the title. We agree that the title might suggest a causal relationship study, whereas our research primarily investigates factors that may influence performance in the Olympic distance triathlon. We have revised the title accordingly to accurately reflect the scope of our study. 

Comment 2: Line 47, there is no logical connection between the sentence “This study also focuses…” and the previous parts of the introduction.

Our response: Thank you for pointing out the coherence issue in the introduction. We acknowledge the issue highlighted regarding the lack of logical connection in the sentence "This study also focuses..." on line 47. We have thoroughly revised the introduction, and the sentence in question has been removed entirely.

Comment 3: The introduction does not follow a logical and coherent approach to introduce the main issue, present the current knowledge, indicate the gaps and novel aspects of the study, and finally form a concrete hypotheses or questions. I highly recommend the authors to reconstruct the introduction.

Our response: Thank you for your feedback. We appreciate your thorough review of the introduction. We have taken your suggestions into consideration and work on reconstructing the introduction to ensure it follows a logical and coherent approach. We have understood the importance of properly introducing the main issue, presenting current knowledge, indicating gaps and novel aspects of the study. 

Comment 4: Line 95, please include the name of the official website form where the environmental data were collected.

Our response: Thank you so much for your careful check. We have indicated the name of the official website where the environmental data was collected on line 95.

Comment 5: Line 106, what the authors means by “inconsistent time record” which was set as an exclusion criterion?

Our response: We thank the reviewer for the very interesting comment. The term "inconsistent time record" refers to cases where there are discrepancies or anomalies in the recorded times of the athletes. This could occur due to various reasons, such as errors in recording the results, issues with data uploading on the website, or other factors that may lead to inconsistencies in the reported times. For example, if the sum of the split times for three segments (swim, bike, and run) and the two transition times (T1 and T2) does not match the total time, or if the individual segment times deviate significantly from expected norms, it suggests that there might be issues with the data for that particular athlete's performance. As a result, these inconsistent time records are excluded from the dataset to ensure the integrity and reliability of the analysis.

Comment 6: In Table 2, there is an inconsistency between total time and the time of each segment (swim, bike, and run).

Our response: We feel sorry for the inconvenience brought to the reviewer. Table 2 provides a descriptive statistic of the data, showing the maximum and minimum values for overall time in seconds, as well as the time for each segment (swim, bike, and run). However, it is important to note that the shortest total time does not necessarily correspond to the lowest time for each individual segment. Similarly, when the swim time is at its minimum, it does not mean that the athlete's overall performance is the best. This is because each segment's time is calculated independently, and the shortest total time simply reflects the combination of those individual times. Therefore, the inconsistency between the sum of segment times and the total time is a normal phenomenon.

Comment 7: Line 108, the Statistical Analysis section must include all the details related to any statistical tests used in the study. There are some missing points that should be considered.

Our response: Thank you so much for your careful check. For the statistical analysis section, we have added all the relevant details related to the statistical tests used in the study, e.g., line 155 and 162. Please let us know if there are any omissions.

Comment 8: Is investigating the difference between two sexes one of the study goals? Nothing has been mentioned in this regard.

Our response: Thank you for your feedback. Investigating the difference between the two sexes is not one of the study goals. However, it's important to address gender differences as they can impact the model's outcomes. If there are substantial performance variations between male and female triathletes, treating them as a single group for quantile regression may not be suitable. Therefore, we investigated the difference between two sexes before proceeding.

Comment 9: While most of the quantile regression studies are interested in finding the median of Y, why the authors set the Ƭ of 0.1th and 0.9th in the present study?

Our response: Thank you so much for your careful check. The selection of 0.1 and 0.9 quantiles instead of focusing solely on the median is based on the intention to capture extreme performance outcomes and assess the impact of factors at the lower and upper ends of the performance spectrum. Typically, most quantile regression studies focus on identifying the median of the dependent variable Y, which corresponds to τ=0.5. However, in the present study, set τ at 0.1 and 0.9 because we were interested in performance between top performers (top 10%) and average performers (top 90%, representing the majority of normal athletes) concerning the three transition times and environmental factors such as temperature, altitude, and water conditions.

Comment 10: In the results section, more detailed information regarding the order of influence of each independent variables for each quintile and also for sexes must be provided. All these information is missing in the results section. In the meantime, the authors have used such information in the discussion section without presenting them in the result section.

Our response: Thank you so much for your careful check. We have provided a detailed presentation of quantile regression results in the Results section, exemplifying the interpretation of findings using one specific quantile. In the Discussion section, we have meticulously analyzed and discussed the results in alignment with our findings.

Comment 11: In the discussion section, there are no strong support for the findings of the study. For example, the authors arguing that by increasing the altitude the performance increases and attributed this finding to acclimatization of the athletes. However, the mean altitude of races was 186 m above the sea level which normally has no considerable effect of body. In another example, the authors stated that the difference in the importance of running and swing as the predictors of the overall time is attributed to the difference in physical activity of athletes at 0.1 and 0.9 percentile. Why is that? No further information has been provided.

Our response: Thank you so much for your careful check. We have included more analysis and relevant literature support in the discussion section. All the relevant contents you have pointed out have been modified, specifically on the lines 308-318.

Comment 12: The main text of the manuscript needs a very careful revi

---

## [Decision Letter · Decision Letter 1]

23 Jul 2024

PONE-D-24-09996R1Analysis of quantile regression for Performance in Olympic Distance TriathlonsPLOS ONE

Dear Dr. Zhao,

Thank you for submitting your manuscript to PLOS ONE. After careful consideration, we feel that it has merit but does not fully meet PLOS ONE’s publication criteria as it currently stands. Therefore, we invite you to submit a revised version of the manuscript that addresses the points raised during the review process. **While the reviewers were generally favorable, I strongly encourage you to carefully consider and address the concerns of reviewer 1. The reviewer has offered some good suggestions on how to address the deficiencies outlined. In your revisions and response please specifically identify how you have addressed these concerns with a justification for your approach.**

We look forward to receiving your revised manuscript.

Kind regards,

Chris Harnish, PhD

Academic Editor

PLOS ONE

Reviewers' comments:

Reviewer's Responses to Questions

**Comments to the Author**

1. If the authors have adequately addressed your comments raised in a previous round of review and you feel that this manuscript is now acceptable for publication, you may indicate that here to bypass the “Comments to the Author” section, enter your conflict of interest statement in the “Confidential to Editor” section, and submit your "Accept" recommendation.

Reviewer #1: (No Response)

Reviewer #2: All comments have been addressed

2. Is the manuscript technically sound, and do the data support the conclusions?

Reviewer #1: Partly

Reviewer #2: Yes

3. Has the statistical analysis been performed appropriately and rigorously? 

Reviewer #1: No

Reviewer #2: Yes

4. Have the authors made all data underlying the findings in their manuscript fully available?

Reviewer #1: Yes

Reviewer #2: Yes

5. Is the manuscript presented in an intelligible fashion and written in standard English?

Reviewer #1: Yes

Reviewer #2: Yes

6. Review Comments to the Author

**Reviewer #1:** The authors have addressed many of the issues but the largest issue, unless i've somehow got the wrong version, that they regress total time on the sum of swim, bike and run times, and then, not surprisingly get coefficients near one, and then interpret say, 'swim' as 'more important has not been addressed. Very generally, if Y = A + B + C, and you regress Y on A, B, and C, and then get Y = aA + bB + cC and have found a=1, b=1, c=1, you have a regression that isn't telling you any at best; at worst, OLS will not run as there is no error. Equation 1 is of this form, and then the results of table 5 confirm the coefficient estimates =1. There are many ways to deal with this, one being instrumental variables, or better still use 3 equations -- but one equation should be dropped -- i.e., total, swim, bike and run times, there are only 3 exogenous equations out of 4. This is the elephant in the room. Another way of dealing with this is to create 'share' equations -- .e.g, share of the total -- there is still one equation endogenous with shares very generally note. There are then other things, such as whether male and female can/or should be pooled, interacting this or not, etc.

**Reviewer #2: **All of the points that I raised in my previous comments have now been addressed. I have no further concerns and therefore consider the manuscript to be acceptable in its current format.

7. PLOS authors have the option to publish the peer review history of their article (what does this mean?). If published, this will include your full peer review and any attached files.

Reviewer #1: No

Reviewer #2: No

---

## [Author Response · Author response to Decision Letter 1]

3 Sep 2024

In response to Reviewer #1's comments and the modifications made, we provide the following clarification:

We have revised the model as detailed in lines 141-156 of the manuscript. The model results are on lines 181-189. Specifically:

1、Overall time (seconds) = Swim time (seconds) + Bike time (seconds) + Run time (seconds) + Transition 1 + Transition 2. Since transition times are not the focus of this study, they are not analyzed in the manuscript.

2、Given the correlation coefficient of 0.55 between Run time and Bike time, we have excluded Run time from the quantile regression model.

3、We included all performance data and introduced a gender dummy variable to expand the sample size, enhance the robustness of our results, and examine the effect of gender on triathlon performance.

These changes address the issue of regressing overall time on the sum of swim, bike, and run times, thus avoiding coefficients close to 1. The discussion and conclusions of the revised model have been updated accordingly, as noted in lines 230, 257, 313, and elsewhere.

Regarding the use of instrumental variables, we attempted to use the water environment as an instrument for swim time. The water environment theoretically meets the requirements for an instrumental variable for swim performance (since it influences swim performance but does not directly affect the overall time). However, using the water environment alone did not yield satisfactory results, and the model performance was poor. Therefore, this approach was not adopted.

 Reviewer’s points are valuable. We acknowledge that more data on suitable instrumental variables (e.g., sub-event training performance, physical tests, training duration) would be beneficial. However, we cannot access this data in the short term. We have discussed this limitation in the Discussion section and will consider it seriously for future research.

We hope that the manuscript, after careful revisions, meets our high standards. The authors welcome additional constructive comments, if there are any. Thank you once again for your guidance throughout the review process. We look forward to hearing from you regarding the next steps.

---

## [Decision Letter · Decision Letter 2]

17 Sep 2024

PONE-D-24-09996R2Analysis of quantile regression for Performance in Olympic Distance TriathlonsPLOS ONE

Dear Dr. Zhao,

Thank you for submitting your manuscript to PLOS ONE. After careful consideration, we feel that it has merit but does not fully meet PLOS ONE’s publication criteria as it currently stands. Therefore, we invite you to submit a revised version of the manuscript that addresses the points raised during the review process.

We look forward to receiving your revised manuscript.

Kind regards,

Przemysław Seweryn Kasiak

Academic Editor

PLOS ONE

**Additional Editor Comments:**

Dear Authors, your manuscript has been checked by the experts in this field. Please address all the comments from Reviewer #3 before final decision.

Reviewers' comments:

Reviewer's Responses to Questions

**Comments to the Author**

1. If the authors have adequately addressed your comments raised in a previous round of review and you feel that this manuscript is now acceptable for publication, you may indicate that here to bypass the “Comments to the Author” section, enter your conflict of interest statement in the “Confidential to Editor” section, and submit your "Accept" recommendation.

Reviewer #2: All comments have been addressed

Reviewer #3: (No Response)

2. Is the manuscript technically sound, and do the data support the conclusions?

Reviewer #2: Yes

Reviewer #3: Partly

3. Has the statistical analysis been performed appropriately and rigorously? 

Reviewer #2: Yes

Reviewer #3: I Don't Know

4. Have the authors made all data underlying the findings in their manuscript fully available?

Reviewer #2: Yes

Reviewer #3: Yes

5. Is the manuscript presented in an intelligible fashion and written in standard English?

Reviewer #2: Yes

Reviewer #3: Yes

6. Review Comments to the Author

**Reviewer #2:** (No Response)

**Reviewer #3:** Thank you for providing an opportunity to review the manuscript. The general idea is interesting and the presented results seem worth publishing and discussing. To some extent, this is the complementary paper for “A study of triathletes’ race strategies in different competition environments” published in Heliyon 2024, by the same first author. However, revisions are required. Most importantly, I believe the necessary improvements are doable, therefore I encourage the authors to put the work into improving the paper. Please find my suggestions below:

1) You cannot phrase that favorable river current or altitude improves performance - the times might be faster, but it does not mean that the performance was better. Certain conditions might have resulted in faster times, and that is what you have analyzed in terms of the environment. Consequently, rephrase the goal of the study in lines 68-70 and all the relevant sections in the paper. It will also affect some parts reg interpretation. Faster time does not equal better performance.

2) As you only examined well-trained triathletes (Continental Cups, World Cups, domestic highest-level races) with professional backgrounds why do you differentiate amateurs and professionals? They are all pros, however on different levels. See McKay et al 2022, Participants Classification Framework (DOI: 10.1123/ijspp.2021-0451) and apply it correctly.

3) What about wetsuit influence? With olympic distance (now it is officially called standard distance so consider changing the phrase…) I believe above 20 Celsius wetsuits are forbidden, see World Triathlon Competition Rules for clarification. This is an important factor and should be mentioned. Hopefully, it could be taken into account in the analysis, as the water temp is provided on triathlon.org, at least in the limitations section.

4) Were these draft legal or non-drafting races? Were the competition rules the same for international and domestic races? If yes, mention that.

5) The paper structure is not appropriate, the headlines are not prioritized correctly, and some paragraphs belong to other sections. Ie lines 110-115 are redundant here, if you want to state the focus of the study then do it in the introduction. The methods section should state the design of the study first (retrospective analysis of public data) and then, dig into the details.

In multiple places, you do not provide references and you state conclusions (in the abstract all the sections should be changed) that do not originate from the study results or references work.

6) I cannot agree with excluding run split from the analysis due to so-called multicollinearity. A correlation of 0.545 is (conventional approach) considered moderate and without taking run into consideration the results are not really useful.

7) You repeat the results multiple times in the discussion, this is not a goal of the discussion section. Refer to relevant work and provide context for the results that you have already provided in the Results section. If you feel your result presentation is not clear see APA or AMA guidelines, for example.

8) If you underline the role of specific training and testing to the race demands you might refer to the following papers, dois below:

10.23736/S0022-4707.24.15921-X

10.1111/j.1468-2419.2007.00286.x

doi.org/10.3390/sports7050101

10.1123/ijsnem.2018-0256

9) I recommend proofreading with a native speaker or editor. Language needs some improvement in flow and readability. Even the current title is not grammatically correct, with no clear reason why some words begin with capitals or small letters.

Additionally, some minor suggestions:

Lines 39-45 - remove ie from brackets, redundant.

48 - reference needed

Table 1 title - It is not a race schedule, these are events included in the study…

227-228 - speculative, provide reference or delete

270-275 - speculative, provide reference or delete

276-277 - reference needed

284-287 - out of scope, maybe you might address air density and drag as a contribution towards faster overall splits? or no clear explanation?

298 - underline it is a singular example

303 - provide reference

306 - provide reference

307-313 - the whole paragraph is out of the scope of the study

326 - LimitationS

347-351 - redundant

353-358 - speculative and not originating from the results, rephrase or delete

multiple double spacing and occasional capital letters in the middle of the sentence

TO SUM UP: The manuscript requires major revisions to be considered for publication. However, as the study design is acceptable, the results are interesting, the topic is important, all the necessary improvements are possible.

7. PLOS authors have the option to publish the peer review history of their article (what does this mean?). If published, this will include your full peer review and any attached files.

Reviewer #2: No

Reviewer #3: No

---

## [Author Response · Author response to Decision Letter 2]

23 Oct 2024

Dear Chris Harnish,

We are very grateful to you and the reviewers for your positive comments on the study and for the constructive comments you provided. We have carefully considered the suggestions of Reviewer #3 and have made the following revisions. All revisions are highlighted in blue color in the manuscript.

In response to Reviewer #3's comments and the modifications made, we provide the following clarification:

1) You cannot phrase that favorable river current or altitude improves performance - the times might be faster, but it does not mean that the performance was better. Certain conditions might have resulted in faster times, and that is what you have analyzed in terms of the environment. Consequently, rephrase the goal of the study in lines 68-70 and all the relevant sections in the paper. It will also affect some parts reg interpretation. Faster time does not equal better performance.

Our response: Thank you for your insightful feedback regarding the interpretation of performance in relation to environmental factors. We appreciate your clarification that faster times do not necessarily equate to better performance, as they may be influenced by favorable conditions such as river currents or altitude.

In response to your comments, we will revise the goal of the study stated in lines 83-92 and adjust all relevant sections throughout the paper to reflect this important distinction. We will emphasize that our analysis focuses on how specific environmental conditions may impact the recorded times without making direct claims about improvements in athletic performance itself. Additionally, we recognize that this adjustment will affect various parts of the interpretation in the paper, and we will ensure that our conclusions are consistent with this updated perspective.

2) As you only examined well-trained triathletes (Continental Cups, World Cups, domestic highest-level races) with professional backgrounds why do you differentiate amateurs and professionals? They are all pros, however on different levels. See McKay et al 2022, Participants Classification Framework (DOI: 10.1123/ijspp.2021-0451) and apply it correctly.

Our response: Thank you for your insightful feedback regarding the differentiation between amateurs and professionals in our study. Indeed, we examined well-trained triathletes (Continental Cups, World Cups, and domestic highest-level races) with professional backgrounds.

In our study, we focused on the performance records of the "Elite Group," as classified by the China Triathlon Association, which includes categories such as Elite, Youth, and U19. The emphasis on the "Elite Group" is due to the highly competitive ability of the athletes within this category, which provides more representative data for our analysis. This group comprises both male and female elite athletes, corresponding to the Elite Men and Elite Women categories on the official World Triathlon website. We have accurately described this in the manuscript, specifically in line 101-104 for clarity.

Regarding your reference to the study by McKay et al. (2022), it is important to note that due to the challenges of quantifying performance in triathlon, the original paper states: "In sports where quantifying performance isn’t appropriate (e.g., rowing, where environmental conditions and wind speeds can affect performance; and BMX/Mountain biking where races are completed on different courses), athlete ranking and placings at major competitions should be the priority metric used to classify athletes." Therefore, we have aligned our classification with the World Triathlon, which define the Elite Group as a professional triathlete who competes at an international level.

3) What about wetsuit influence? With olympic distance (now it is officially called standard distance so consider changing the phrase…) I believe above 20 Celsius wetsuits are forbidden, see World Triathlon Competition Rules for clarification. This is an important factor and should be mentioned. Hopefully, it could be taken into account in the analysis, as the water temp is provided on triathlon.org, at least in the limitations section.

Our response: Thank you for your valuable feedback regarding the influence of water temperature and wetsuits on the analysis of the triathlon events. We appreciate your insights into this important aspect.

In our study, we analyzed triathlon competitions held in China from 2013 to 2023, including the National Triathlon Championship Cup series. However, we found that many events prior to 2020 did not have recorded water temperature data on the China Triathlon Association website (http://triathlon.sport.org.cn/china/), which restricts our ability to analyze the effects of water temperature and the use of wetsuits.

We have clearly pointed out this limitation in the limitations section of our paper, demonstrating our commitment to data integrity and the rigor of our analysis. Furthermore, we plan to include the water temperature variable in future studies and will seriously consider its impact on competition results.

4) Were these draft legal or non-drafting races? Were the competition rules the same for international and domestic races? If yes, mention that.

Our response: Thank you for your valuable feedback! In the standard distance triathlon "Elite Group" races, drafting is allowed. This rule is consistent with the World Triathlon regulations, as specified in our manuscript on line 109-112.

Additionally, the competition rules are the same for both international and domestic races, ensuring consistency in how events are conducted. For further details, you can refer to the World Triathlon Competition Rules at the following link: [World Triathlon Competition Rules] (https://www.triathlon.org/uploads/docs/World_Triathlon_Sport_Competition_Rules_2020_201811253.pdf#:~:text=The World Triathlon Competition Rules are intended to: (i).

5) The paper structure is not appropriate, the headlines are not prioritized correctly, and some paragraphs belong to other sections. Ie lines 110-115 are redundant here, if you want to state the focus of the study then do it in the introduction. The methods section should state the design of the study first (retrospective analysis of public data) and then, dig into the details.

Our response: Thank you for your detailed feedback regarding the structure of the paper. We have adjusted the prioritization of the headings. Additionally, we have removed the redundant content in lines 110-115 as you suggested.

Regarding the missing references, we have carefully reviewed our conclusions to ensure that all statements are adequately supported by relevant literature. This can be found in the revised manuscript on lines 254, 257, and 299.

6) I cannot agree with excluding run split from the analysis due to so-called multicollinearity. A correlation of 0.545 is (conventional approach) considered moderate and without taking run into consideration the results are not really useful.

Our response: We appreciate your comments on the exclusion of the run split analysis, particularly considering that a correlation of 0.545 is deemed moderate. We have re-evaluated this decision and have decided to retain the run split in the regression analysis to enhance the practical utility of our results. The updated results can be found on line 179-182.

7) You repeat the results multiple times in the discussion, this is not a goal of the discussion section. Refer to relevant work and provide context for the results that you have already provided in the Results section. If you feel your result presentation is not clear see APA or AMA guidelines, for example.

Our response: The discussion has been carefully reviewed to eliminate any repetitive descriptions of the results. Relevant literature has been cited to provide additional context for the findings.

8) If you underline the role of specific training and testing to the race demands you might refer to the following papers, dois below:

10.23736/S0022-4707.24.15921-X

10.1111/j.1468-2419.2007.00286.x

doi.org/10.3390/sports7050101

10.1123/ijsnem.2018-0256

Our response: Thank you for your valuable feedback regarding the references to specific training and testing in relation to race demands. We appreciate your suggestions and have incorporated them into the manuscript. In the Introduction, specifically lines 60-72, we have now included the references to underline the importance of tailored training and testing protocols in addressing the demands of racing.

9) I recommend proofreading with a native speaker or editor. Language needs some improvement in flow and readability. Even the current title is not grammatically correct, with no clear reason why some words begin with capitals or small letters.

Our response: The manuscript has been thoroughly proofread to enhance flow and readability. Additionally, the title has been revised to ensure it adheres to grammatical standards.

We would like to express our sincere gratitude for the reviewer's helpful suggestions. We have carefully reviewed and made the necessary revisions, which are detailed below:

1. Lines 39-45: The term "i.e." has been removed as it was deemed redundant.

2. Line 48: A relevant reference has been added to support the statement.

3. Table 1 Title: The title has been revised to clarify that it lists events included in the study rather than a race schedule.

4. Lines 227-228 and 270-275: The speculative content has been deleted.

5. Lines 276-277: The content requiring a reference has been deleted in the revision.

6. Lines 284-287: This section, deemed out of scope, has been removed.

7. Line 298: The content has been deleted.

8. Lines 303 and 306: Both sentences requiring references have been removed from the manuscript.

9. Lines 307-313: This paragraph has been removed as it was out of the scope of the study.

10. Line 326: The typo in "LimitationS" has been corrected to "Limitations."

11. Lines 347-351: Redundant content has been deleted for conciseness.

12. Lines 353-358: Speculative content that did not originate from the results has been deleted.

We believe these modifications enhance the clarity and rigor of the manuscript. Thank you once again for your valuable suggestions, which have greatly contributed to our work. If further adjustments or additions are needed, please let us know.

Lastly, we hope that the manuscript, after careful revisions, meets our high standards. The authors welcome additional constructive comments, if there are any. Thank you once again for your guidance throughout the review process. We look forward to hearing from you regarding the next steps.

Sincerely,

The Authors

---

## [Editor Report · Decision Letter 3]

25 Oct 2024

Analysis of quantile regression for race time in standard distance triathlons

PONE-D-24-09996R3

Dear Dr. Zhao,

We’re pleased to inform you that your manuscript has been judged scientifically suitable for publication and will be formally accepted for publication once it meets all outstanding technical requirements.

Kind regards,

Przemysław Seweryn Kasiak

Academic Editor

PLOS ONE
---

## [Editor Report · Acceptance letter]

14 Nov 2024

PONE-D-24-09996R3 

PLOS ONE

Dear Dr. Zhao, 

I'm pleased to inform you that your manuscript has been deemed suitable for publication in PLOS ONE. Congratulations! Your manuscript is now being handed over to our production team.

Kind regards, 

on behalf of

Dr. Przemysław Seweryn Kasiak 

Academic Editor

PLOS ONE